# Differences in Cytokine Expression at Baseline and in Response to Mineral Stimulation by Peripheral Blood Mononuclear Cells from Podoconiosis Cases and Healthy Control Individuals

**DOI:** 10.3390/tropicalmed9110252

**Published:** 2024-10-22

**Authors:** Mikias Negash, Tigist Girma, Menberework Chanyalew, Dawit H. Alemayehu, Diana Alcantara, Gail Davey, Rosemary J. Boyton, Daniel M. Altmann, Melanie J. Newport, Rawleigh Howe

**Affiliations:** 1Brighton and Sussex Centre for Global Health Research, Department of Global Health and Infection, Brighton and Sussex Medical School, Brighton BN1 9PX, UK; 2Armauer Hansen Research Institute, Addis Ababa P.O. Box 1005, Ethiopiamenberework@gmail.com (M.C.);; 3Department of Medical Laboratory Science, College of Health Sciences, Addis Ababa University, Addis Ababa P.O. Box 1176, Ethiopia; 4School of Public Health, Addis Ababa University, Addis Ababa P.O. Box 1176, Ethiopia; 5Department of Infectious Disease, Imperial College London, London W2 1PG, UK; 6Department of Immunology and Inflammation, Imperial College London, London W12 0NN, UK

**Keywords:** podoconiosis, non-filarial lymphoedema, pro-inflammatory cytokines, gene expression, mineral stimulation

## Abstract

Epidemiological, histological, and immunogenetic studies suggest that podoconiosis (a non-infectious tropical lymphoedema affecting approximately 4 million people globally) is an HLA class II-associated inflammatory condition that develops in response to an unknown trigger found in volcanic red clay soils. Silicate particles of the kaolinite and aluminum types have been identified in femoral lymph node biopsy samples from endemic area residents, suggesting a possible role in the pathogenesis of podoconiosis. We measured in vitro peripheral blood mononuclear cell cytokine responses (*TNF-α*, *IL-1β*, and *IFN-γ*) to stimulation with the minerals kaolinite, chlorite, and beryllium sulfate (all at 100 µM) using ELISA. Real time PCR was used to measure gene expression of signature cytokines in fresh whole blood, comparing podoconiosis patients and endemic healthy controls. Our results showed that the levels of *TNF-α* and *IL-1β* from in vitro cell cultures were significantly higher in unstimulated samples from patients compared to controls (*p* = 0.04 and *p* = 0.005, respectively). The minerals kaolinite and chlorite induced two and three-fold higher levels of *IL-1β* following 24 h of stimulation in healthy controls compared to patients, respectively. We did not find significant differences in mRNA expression of the cytokine genes assayed, though a slight fold increment in *IL-1β* and *TGF-β* was observed. In conclusion, our data suggest that the immune system is in a state of persistent activation in vivo in podoconiosis patients, and additional studies of immune regulation and exhaustion are needed to further characterize immune dysfunction in the pathogenesis of the disease. A better understanding of the underlying processes could lead to the development of a ‘biosignature’ detectable in the early reversible stages that could ultimately contribute to the elimination of this preventable, disabling, neglected tropical disease.

## 1. Introduction

Podoconiosis (non-filarial lymphoedema) causes progressive swelling of the lower legs in individuals who walk barefoot on red clay soils of volcanic origin. The disease is endemic in tropical countries in areas with an altitude of more than 1000 m above sea level and annual rainfall of more than 1500 mm [1]. African areas where podoconiosis is endemic share similar features in terms of climatic, geological, and soil conditions, with a predominance of aluminum and silicon minerals in the soil [2,3,4,5,6].

The epidemiological association between disease prevalence and volcanic red soil led to the suggestion that the disease is induced by mineral particles found in the soil [7,8,9]. Electron microscopy-based elemental analysis of lymph nodes collected from the inguinal sites of people with and without lymphoedema was carried out by Dr Ernest Price in the 1970’s. Silica particles were found in the lymph node macrophages of both groups, although birefringent (uncoated or bare silica) particles were substantially more common in those with lymphoedema [9,10]. The predominant silica particles were mainly of the kaolinite type, sized between 0.2 and 2 µm [11]. A study of soil content in Ocholo, in the Rift Valley region in southeast Ethiopia, indicated that levels of minerals found predominantly in basaltic rocks, including trace elements like beryllium and zirconium, were much higher in the soil from podoconiosis endemic areas compared to neighboring areas where podoconiosis prevalence was low [12]. Fromell et al., who carried out this analysis, suggested the mineral particles in volcanic soil could induce granuloma formation in the lymph node and lead to lymph node sclerosis and eventually to lymphoedema [12]. The elemental content from these microanalyses of lymph node samples showed correlation with elements and minerals derived from volcanic rocks in podoconiosis endemic areas [1,5,6,13].

Price also demonstrated collagenization and obliteration of the lymphatic lumen with infiltrates of lymphocytes and macrophages in lymph node biopsy samples from podoconiosis patients suggestive of an inflammatory process [9]. A more recent study that assessed the histopathological and immunohistochemical features of nodular podoconiosis indicated that patients had thickened dermal collagen, reduced elastic fibers, dilated and often sclerotic blood vessels, a moderate lymphoplasmacytic infiltrate with mast cells, and scattered macrophages [14].

Not everyone exposed to endemic soil develops podoconiosis, and there is a reproducible association with variation in genes located in the HLA class II region [15,16]. This suggests a central role for cell-mediated immune responses in the pathogenesis of podoconiosis. Studies to further investigate the immunological basis of podoconiosis are limited by the lack of knowledge about the soil component that drives the immune response. Minerals present in soil from an endemic region are candidate ‘antigens’, given that there are other examples of HLA-associated diseases triggered by minerals such as silicosis and berylliosis, and mineral deposits are present in individuals’ lymph nodes in podoconiosis-endemic regions. This study, therefore, aimed to characterize the inflammatory response in podoconiosis through measurement of proinflammatory cytokine levels both in the ‘innate state’ using unstimulated peripheral blood and also in response to in vitro stimulation of peripheral blood mononuclear cells (PBMCs) with potentially pathogenic minerals. Real time-PCR was performed using peripheral blood to measure signature cytokines representing the different immune response pathways (*IFN-γ*, *TNF-α*, *TGF-β*, *IL-10*, *IL-1β*, and *IL-4*). This is the first time such in vitro stimulation studies have been conducted in podoconiosis. This approach may advance understanding of the immunological basis of the disease, help identify the soil trigger through a reverse immunology approach, and aid the development of a point-of-care test for early case detection, ultimately enabling elimination.

## 2. Results

The overnight incubation was designed to target pro-inflammatory cytokines produced mainly by monocytes, dendritic cells, and other myeloid lineage cells, and the longer-term cultures to target the T cell-produced *IFN-γ*, allowing time-augmented responses related to T cell proliferation.

### 2.1. TNF-α and IL-1β Levels Following 24 h In Vitro Mineral Stimulation

Our result from the overnight stimulation experiment showed that *TNF-α* levels were elevated in the supernatants from podoconiosis patients in unstimulated cultures compared to healthy controls, with a median value of 783.6 pg/mL vs. 445 pg/mL, respectively (*p* = 0.04). There were no significant differences in *TNF-α* levels between the two groups following overnight stimulation with any of the three minerals tested, kaolinite, chlorite, and BeSO_4_ (Figure 1A, Figure 1B, and Figure 1C, respectively). Similarly, the *IL-1β* levels were significantly higher in podoconiosis patients in the unstimulated wells compared to healthy controls (*p* = 0.005, with a median value of 445 pg/mL vs. 237 pg/mL). Moreover, kaolinite and chlorite induced significantly higher *IL-1β* responses compared to the unstimulated control cultures in both groups. The augmentation was substantially higher in healthy controls compared to podoconiosis patients, with kaolinite showing two-fold enhanced production over unstimulated levels (*p* < 0.001) and chlorite exhibiting three-fold higher levels (*p* < 0.0001) (Figure 1D,E). On the other hand, BeSO_4_ inhibited these cytokines to levels lower than those of unstimulated culture wells for both study groups (Figure 1C,F).

### 2.2. IFN-γ Levels in Response to Mineral Stimulation Following PHA Priming

In this approach, we evaluated the modulatory effect of minerals on the T cell *IFN-γ* response to the polyclonal stimulus PHA. The levels of *IFN-γ* were significantly higher in cultures from podoconiosis patients compared to healthy controls in the presence of the mineral kaolinite, with a median value of 2491 pg/mL vs. 384 pg/mL, respectively, *p* = 0.001, and chlorite, with a median value of 1990 pg/mL vs. 364.2 pg/mL, *p* = 0.006 (Figure 2A and Figure 2B, respectively). No difference was observed for PHA pulsed and BeSO_4_ stimulated wells, Figure 2C. However, *IFN-γ* levels from podoconiosis and healthy control supernatants pulsed with PHA and stimulated with minerals were both lower than the levels observed in positive control cultures (PHA only). So in effect, the minerals had a suppressive effect on the responses to PHA stimulation after priming with PHA, which was more pronounced in cells from healthy controls than it was in podoconiosis patients.

It was further observed that the median mineral-induced *IFN-γ* level decreased by 75% for kaolinite and by 78% for chlorite relative to the PHA-only positive control in healthy controls (Figure 3A, Figure 3B, and Figure 3C, respectively). On the other hand, BeSO_4_-induced suppression was comparable between the two study groups (Figure 3C).

### 2.3. Cytokine Gene Expression in Peripheral Blood from Podoconiosis Patients and Healthy Control Subjects

The expression of mRNA for selected cytokine genes (*IL-1β*, *IL-4*, *IL-10*, *TNF-α*, *IFN-γ*, and *TGF-β*) in unstimulated peripheral blood samples was measured using a two-step qRT-PCR assay. The relative mRNA level (gene expression) of *IL-1β* in podoconiosis patients was 1.8-fold higher than in healthy controls, although it did not reach a statistically significant level (*p* = 0.27, Figure 4A). Similarly, the expression of *TGF-β* was 1.5 times higher in podoconiosis patients compared to healthy controls (*p* = 0.28, Figure 4F). On the other hand, the expression of *IL-4* mRNA level was 1.8-fold higher in healthy controls compared to podoconiosis patients (*p* = 0.27, Figure 4B). There was no difference in the expression of the remaining cytokines, *IL-10*, *TNF-α*, and *IFN-γ*, between the patients and the healthy controls (Figure 4C–E).

## 3. Discussion

Podoconiosis is a form of tropical lymphoedema that has a strong genetic association with variation at class II HLA loci, and its development requires exposure to volcanic clay soils with specific mineral compositions. Given these findings, we hypothesized that immune activation and inflammation triggered by mineral particles play a key role in the pathogenesis of podoconiosis. In the current study, we explored peripheral blood cytokines levels and PBMC response to kaolinite, chlorite, or beryllium in podoconiosis patients and healthy controls.

*TNF-α* is an important pro-inflammatory cytokine with pleiotropic actions on cells, promoting host defense, cell proliferation, and inflammation. However, inappropriate or excessive activation of *TNF-α* signaling is associated with chronic inflammation, which, if persistent and unregulated in susceptible hosts, can exacerbate tissue injury in different inflammatory and autoimmune diseases [17]. Averting the persistence of *TNF-α*-associated inflammation and restoring immune regulation has been the main target of biological therapies, whereby *TNF-α* blockade using monoclonal antibodies has been widely used to treat autoimmune diseases such as rheumatoid arthritis [18,19] and ulcerative colitis [20], and this could potentially be useful in reducing inflammation in podoconiosis, though a lot more research is required before reaching this step as the increase in *TNF-α*- in the patients is only moderately significant.

Similarly, there were significantly higher levels of *IL-1β* measured in the 24 h mineral stimulation assay in podoconiosis patients in unstimulated wells compared to healthy controls, which suggests a state of inflammation in vivo. *IL-1β* is also a prominent pro-inflammatory cytokine whose aberrant expression and regulation drives disease progression in different chronic inflammatory conditions, such as type 2 diabetes [21], and chronic inflammation involving skin and bone [22].

Even though higher levels of the pro-inflammatory cytokines *TNF-α* and *IL-1β* were measured from unstimulated cultures in the patient samples, the level of these cytokines increased significantly in healthy control samples after 24 h of stimulation for both cytokines, in particular for *IL-1β*. Given that minerals may trigger inflammation in podoconiosis, it was anticipated that podoconiosis patients would have a more pronounced response to mineral stimulation compared to the healthy controls. However, podoconiosis patients already had significantly higher levels of these cytokines in unstimulated wells compared to the healthy control samples, and while they also responded to mineral stimulation, the fold increase in cytokine levels relative to the baseline levels was lower than the healthy control levels. Despite differences in response to mineral stimulation, the final levels after stimulation were not significantly different between the two study groups. This suggests that the immune system in podoconiosis cases was already partially activated and the control responses quickly ‘caught up’. This lower percentage increase in cytokine levels between baseline and after stimulation in the cases may be secondary to persistent stimulation and activation in vivo, which could lead to exhaustion. It is well documented that persistent stimulation of the immune system induces immune exhaustion in different chronic diseases, including tuberculosis, HIV, and patients with filarial lymphedema [23]. Previous experiments in healthy control individuals by Dillingh et al. showed that in vivo administration of lipopolysaccharide (LPS) dose dependently resulted in hypo-responsiveness during subsequent ex vivo LPS stimulation as measured by low levels of *TNF-α* and *IL-1β* cytokines compared to the placebo groups [24]. Therefore, it is worth exploring immune exhaustion in podoconiosis patients relative to controls in future studies.

In vitro experiments by Franchi et al. with prior priming of monocytes and dendritic cells with microbes, LPS, and *TNF-α* followed by administration of a second signal such as ATP or silicate particles demonstrated that persistent production of *IL-1β* occurred only in those primed first with *TNF-α*, where higher levels were detected up to 5 days post stimulation. Meanwhile, the *IL-1β* levels were shown to wane after 24 h of stimulation in those stimulated with LPS or microbes. The exact mechanism of the activation was not clear, but accumulating evidence indicates this two-step model of activation is mediated by the generation of inflammasomes via the Nucleotide-Binding Domain and Leucine-rich Repeat Containing Protein (NLRP3) pathway, which activates caspase-1, eventually cleaving pro-*IL-1β* to active *IL-1β* [25]. Hence, this suggests that the two cytokines (*TNF-α* and *IL-1β*) may work synergistically in driving the inflammatory process in podoconiosis patients. We have previously demonstrated higher expression of activation markers on classical monocytes and myeloid dendritic cells in podoconiosis patients [26]. These two innate cells are the predominant sources of pro-inflammatory cytokines, and the higher levels of *TNF-α* and *IL-1β* in podoconiosis patients may have resulted from these activated phagocytic cells. Future studies from biopsy samples targeting caspase-1 and NLRP3 levels are important to verify if these pathways in deed could play a role in podoconiosis disease.

The PHA priming assay revealed that the levels of *IFN-γ* in podoconiosis patients after mineral stimulation were comparable to their responses to PHA alone (positive control wells) at day 6, while the healthy controls had significantly reduced levels to PHA and mineral stimulation, suggesting that PHA responses to priming were curbed in the healthy controls by the mineral stimulation. How the minerals induced such a response in healthy controls is not clear. In one study, stimulation of human alveolar macrophages derived from bronchoalveolar lavage of healthy donors with silica particles showed that macrophages with a suppressive phenotype (RFD1^+^/7^+^) were low in number while those with an activator or antigen presenting phenotype (RFD1^+^/7^−^) were higher. It was suggested the lack of immune regulation or suppression allowed an uncontrolled response of the activator phenotype [27,28]. It should be noted that the stimulation assay was developed and optimized by our group in the absence of established protocols suitable for our context, and further studies are required to replicate these findings.

Our real time PCR assay showed that the level of *IL1-β* mRNA expression was higher in podoconiosis patients relative to healthy controls, perhaps reflecting either greater stability or different regulatory mechanisms compared to *TNF-α* mRNA, given the latter was not detected. Similarly, there was a higher level of *TGF-β* mRNA in podoconiosis patients compared to healthy controls, although it did not reach the statistical significance level. Given that the expression of these genes was measured in peripheral blood, the magnitude of expression could be an underestimate of the level in the affected tissue. This highlights that the inflammatory cytokines and pro-fibrotic transcripts could contribute to disease pathology in podoconiosis patients. For example, significantly increased expression of *TGF-β* and extracellular matrix proteins (such as collagen type I and III) associated with fibrosis has been reported in tissue specimens from breast cancer patients with secondary lymphoedema [29]. There are also a number of other pieces of evidence [29,30] that suggest a key role for *TGF-β* in fibrosis of lymphatic tissue. Studies in mouse models have shown that blocking *TGF-β* markedly decreased tissue fibrosis, increased lymphangiogenesis, and improved lymphatic function, which was attributed to attenuated fibroblast proliferation and decreased extracellular matrix deposition [31].

Overall, we have tried to characterize immune inflammatory cytokine responses to minerals in podoconiosis patients relative to endemic controls. However, it is not clear how silica particles interact with MHC molecules to induce cytokine production. Nevertheless, there is evidence that foreign minerals, particles, or drugs can interact with specific MHC types and induce or modulate immune responses. One example is chronic beryllium disease (CBD), a lung disorder that occurs following long-term exposure to beryllium. The disease is associated with an HLA-DP variant with a specific polymorphism of the beta chain associated with the presence of glutamic acid at position 69. The presence of negatively charged glutamine residue allows beryllium to bind to the HLA-DP molecule, creating a new epitope. This triggers a beryllium-specific polyclonal T cell response leading to inflammation and tissue damage [32].

It is also worth noting that not all individuals with risk-associated HLA haplotypes develop an autoimmune disease or a given condition, indicating other factors also play a role. Environmental factors, the extent and duration of exposure to the trigger, and the type of cytokines produced from activated immune cells likely play a significant role in precipitating the disease and shaping its outcome. Such factors could also add complexity to the development and progress of podoconiosis in susceptible individuals.

## 4. Conclusions

Overall, cytokine assays in response to 24 h mineral stimulation showed no difference in responses between podoconiosis patients and healthy controls, but the patients had a higher level of pro-inflammatory cytokines *IL-1β* and *TNF-α* in unstimulated wells. PHA pulsing of cells followed by mineral stimulation led to significant suppression of *IFN-γ* responses in healthy controls but not in podoconiosis patients, which suggested the patients had a sustained response. Whilst the current study has contributed new insight describing immune responses in podoconiosis through studies on components derived from peripheral blood, there remains a gap regarding the specific cellular interaction with silica particles and immune regulation. Moreover, studies including immunohistochemistry and gene expression characterization of biopsy samples taken from affected tissues could provide a more detailed understanding of the immunological basis of podoconiosis.

## 5. Material and Methods

### 5.1. Study Area and Study Participants

Study participants were recruited from two health centers serving six villages in districts within the West Gojam Zone that were located around Bahir Dar town in the Amhara Regional State (Northwestern Ethiopia). West Gojam Zone covers an area of 1443.37 km^2^. The altitude ranges from 1750 to 2300 m above sea level. The most recent estimate of the total population within the study area was 196,766 people in 2011, and almost all of them were farmers [33]. The geographical topography and the presence of volcanic red clays make the Amhara region endemic for the occurrence of podoconiosis. The prevalence of podoconiosis in the Amhara region was 4%, as reported by Deribe et al. based on nationwide mapping carried out in 2015 [4]. Blood samples were collected from a total of 56 podoconiosis patients and 44 endemic healthy controls. The patients were already diagnosed clinically staged based on the modified Tekola staging system (Figure 5) [34].

The majority of the patients were stage 2 (93%); the mean age with standard deviation of the patients and healthy controls was 45 (10.3) and 37 (9.3) years, respectively. The majority of the patients were male (57%). None of the patients had acute dermatolymphangioadenitis, a painful, acute inflammatory complication of lymphoedema, at the time of enrolment (Table 1). The majority of the healthy control subjects were male (59%), had lived in the same study area as the patients for at least 10 years, did not consistently wear shoes, had no family history of podoconiosis, had no history of chronic disease like HIV, diabetes, liver, kidney, and malignancies, and were in good health on the day of recruitment.

### 5.2. Sample Collection and PBMC Isolation

Peripheral blood was collected from each study participant by venepuncture into heparinized tubes (15 mL) and PAXgene™ Blood RNA Tube (2.5 mL). The heparinized sample was used to isolate PBMCs for the in vitro stimulation experiment, and the PAXgene sample was used for RNA extraction for the RT-PCR assay. PBMCs were isolated from whole blood by density gradient centrifugation using Ficoll (Catalogue no. GE17-5442-03, Sigma, St. Louis, MO, USA).

### 5.3. Cell Culture for In Vitro Mineral Stimulation

Minerals shown to be abundant in podoconiosis endemic soils and that may trigger inflammation [1,5,6,9,10,12,13] were used as stimulants. These included beryllium sulfate (BeSO_4_, Catalogue no. 202789-G), and the phyllosilicate crystal clays, kaolinite (Al_2_Si_2_O_5_(OH)_4_, Catalogue no. 03584-250G) and chlorite ((Mg,Fe)_4_Al_4_Si_2_O_10_(OH)_8_). Minerals were purchased from Sigma and endotoxin tested except the latter, which was kindly donated by the Natural History Museum of London, UK, and which was autoclaved and filtered before use. Briefly, PBMCs (2 × 10^5^ per well) were seeded in 96-well flat bottom sterile culture plates in duplicate. Stock suspensions of powdered minerals were prepared in phosphate buffered saline (PBS) at a concentration of 1000 µM. The minerals were then added to the cells in complete medium at the final concentration of 100 µM. This final concentration was chosen as it had the least cytotoxicity compared to higher concentrations of the minerals in optimization experiments. Moreover, such concentrations have been used in prior experiments for in vitro stimulation. Complete medium (5% normal human serum, Sigma, Catalogue no. H3667-100 mL, in RPMI 1640 medium supplemented with 2 mM L-glutamine, 100 U/mL penicillin, and 100 µg/mL streptomycin) alone was added to wells labeled as background (negative control). Phytohaemaglutinin (PHA) was added at a concentration of 2 µg/mL to positive control wells. For the *TNF-α* and *IL-1β* cytokines assays, PBMCs were stimulated with minerals on the first day; the plates were incubated at 37 °C with 5% CO_2_, and *TNF-α* and *IL-1β* cytokines were assayed after 24 h of stimulation. The six day *IFN-γ* assay was carried out based on a two-step induction approach whereby cells were pulsed with 2 µg/mL PHA on the first day and cultured for 24 h. On the second day the minerals were added, and interleukin-2 (IL-2) at 20 ng/mL was added on the third day. Finally, the supernatant was harvested for the *IFN-γ* assay after six days of stimulation at 37 °C with 5% CO_2_. All supernatants were stored at −80 °C until ELISAs were carried out. (See Table 2 for the time line).

### 5.4. ELISA Assay

Sandwich ELISA kits were used to measure the levels of *IFN-γ* (Catalogue no. DY285B), IL-1^®^ (Catalogue no. DY201), and *TNF-α* (Catalogue no. DY210) from cell culture supernatants. All ELISA kits were purchased from R&D, UK, and the protocols were carried out following the manufacturer’s instructions. In brief, 96-well microtitre plates were coated with the appropriate amount of anti-cytokine capture antibodies and incubated at room temperature overnight. The next day the plates were blocked using 1% BSA in PBS and washed with wash buffer (0.05% Tween 20 in PBS). A total of 100 µL of sample and standard were then added in duplicate to the respective wells. Following two hours of incubation and then washing, captured cytokines were detected by incubating with biotin-labeled detection antibodies. The plate was then washed and incubated with streptavidin-horseradish peroxidase complex. The plate was again washed and a substrate solution was added, followed by incubation. Finally, the reaction was stopped using 1 M H_2_SO_4,_ and optical density was determined using a microplate reader (Emax^R^ Plus, Molecular Devices) at a wavelength set to 450 nm and a correction wavelength of 540 nm.

### 5.5. RNA Extraction and cDNA Synthesis

A two-step quantitative reverse transcriptase PCR was carried out to measure mRNA expression of the housekeeping gene human acidic ribosomal protein (*HuPO*) and of the cytokines *IFN-©*, *TNF-α*, *TGF-^®^*, *IL-10*, *IL-1β*, and *IL-4* from peripheral blood of podoconiosis patients and healthy controls. RNA was extracted from samples collected in PAXgene™ tubes using MagMAX^TM^ bead-based extraction method (Life Technologies, Catalogue no. 4451894). The extracted RNA was checked for quantity and quality using the Qubit^TM^ RNA high sensitivity assay kit (Invitrogen, Catalogue no. Q32855) and the dsDNA high sensitivity assay kit (Invitrogen, Catalogue no. Q32854).

cDNA was synthesized using the QuantiTect Reverse Transcription Kit (Qiagen, Catalogue no. 205311) according to the manufacturer’s instructions. Briefly, gDNA wipeout buffer (2 μL) was added to the sample containing the RNA to remove genomic DNA and RNase-free water was added to make a total volume of 14 μL. Then 6 μL of the master mix (4 μL of RT buffer, 1 μL of primer mix, and 1 μL of reverse transcriptase) was added to the suspension. The overall reaction volume was 20 μL and the thermal cycling conditions for the reaction were set to: 42 °C for 3 min, 42 °C for 20 min, 95 °C for 3 min, and held at 4 °C. The cDNA was used for downstream assays in the two-step real time quantitate PCR.

### 5.6. Real Time Quantitative PCR

The QuantiTect SYBR Green PCR kit (Qiagen, Catalogue no. 204143) was used according to the manufacturer’s instructions for measuring the gene expression profile. Briefly, each reaction mixture contained a total volume of 25 μL comprising 12.5 μL 2X SYBR Green PCR Master Mix, 1 μL forward and 1 μL reverse primers (10 μM), 5.5 μL RNase free H_2_O, and 5 μL of sample cDNA (diluted 1:50). The thermal cycling conditions were set at 95 °C for 15 min, 95 °C for 15 s, 58 °C for 25 s, 25 s at 72 °C, and the cycle was run 40 times. The Bio-Rad Thermocycler 1000^TM^ was used for PCR cycling. The cycle threshold (Ct) value was calculated automatically at the linear phase of the amplification. The primers were designed with Primer3 http://bioinfo.ut.ee/primer3-0.4.0/primer3/ (accessed on 25 March 2022) and verified to yield good amplification products in our lab in previous experiments. All the primers were purchased from Eurofins Genomics, UK. The primer sequences used for RT-PCR are presented in Table 3 below.

### 5.7. Gene Expression and Statistical Analysis

The relative gene expression of each cytokine was normalized to the housekeeping gene *HuPO*, and individual Ct values were used for statistical analysis because the study groups are independent. The 2^−∆Ct^ formula was used to calculate the individual gene expression for each sample [35]. For each cytokine, measurements were removed from the analysis if there was a deviation of more than 0.5 cycles between technical duplicates, leaving 38 podoconiosis and 32 healthy controls in the final analysis. Differential cytokine expression between the patients and the controls was analyzed using a non-parametric Mann–Whitney test. The ratios of gene expression between the patients and the controls were used to calculate the fold change for each cytokine. The efficiency of the PCR experiment was excellent at 110% and a correlation coefficient of 0.99 (see Appendix A).

For the ELISA analysis of the 24 h cytokine response, we used the independent Mann–Whitney U test to compare levels between podoconiosis patients and controls,while the paired Wilcoxon signed-rank test was used to compare baseline (unstimulated wells) values with mineral stimulated wells.

## Figures and Tables

**Figure 1 tropicalmed-09-00252-f001:**
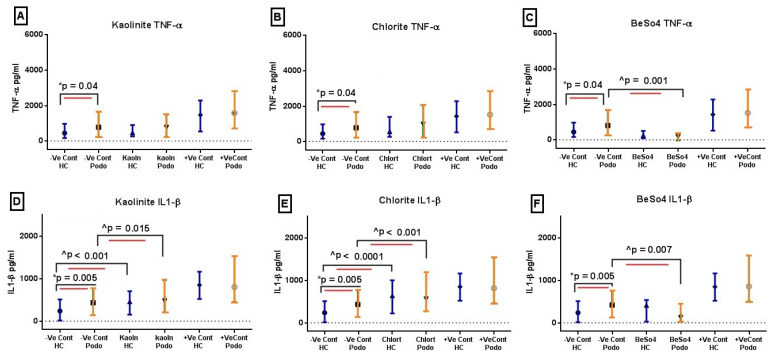
*TNF-α* and *IL-1β* levels in response to 24 h mineral stimulation. ELISA was used to analyze *TNF-α* (upper row plots (**A**–**C**)) and *IL-1β* levels (lower row plots (**D**–**F**)) from supernatants after 24 h of stimulation with 100 µM of kaolinite (**left**), chlorite (**middle**), and BeSO_4_ (**right**). Data for the unstimulated (negative control), mineral-stimulated, and PHA 2 µg/mL (positive control) are depicted here. Each plot depicts *TNF-α* or *IL-1β* from the negative control, mineral-stimulated, and positive control wells for healthy controls (HCs) and podoconiosis patients (Podo). The dots in the middle represent medians, and the whiskers represent inter-quartile ranges from cell culture experiments conducted in samples from 56 podoconiosis patients and 44 healthy controls. Blue vertical bars represent healthy controls and yellow podoconiosis groups. * *p* values were derived using the independent Mann-Whitney U test, and ^ *p* values were derived using the paired Wilcoxon signed-rank test.

**Figure 2 tropicalmed-09-00252-f002:**
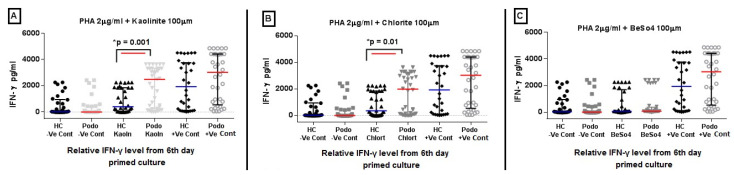
*IFN-γ* levels in response to PHA priming followed by mineral stimulation for 6 days. ELISA was used to analyze *IFN-γ* levels from supernatants after priming cells with 2 µg/mL PHA for 24 h followed by stimulation with 100 µM of kaolinite, chlorite, or BeSO_4_. The supernatant was harvested after 6 days of mineral stimulation (A, B, and C for each mineral, respectively). The plots illustrate the relative expression of *IFN-γ* from unstimulated (negative control), mineral-stimulated, and PHA 2 µg/mL (positive control) wells for healthy control (HC) and podoconiosis patients (Podo) for the minerals (**A**) kaolinite, (**B**) chlorite, and (**C**) BeSO_4_. The colored line in the middle represents the median response, and where appropriate, whiskers represent inter-quartile ranges from 40 podoconiosis patients and 32 healthy controls. *p* values were derived using the Mann–Whitney U test.

**Figure 3 tropicalmed-09-00252-f003:**
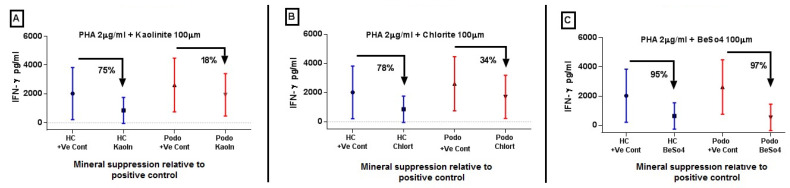
Mineral induced suppression of *IFN-γ* response following PHA priming and stimulation for 6 days. Plots (**A**–**C**) show the degree of suppression induced by 100 µM of kaolinite, chlorite, or BeSO_4_, respectively, relative to the positive control after 6 days of stimulation. PHA pulsed mineral induced suppression is depicted here for kaolinite, chlorite and BeSo4, Figure A, B and C respectively. The blue bar is for HC and the red bar for podo cases. The dots/square/triangle in the middle line represent the median response, and where appropriate, whiskers represent inter-quartile ranges from 40 podoconiosis cases and 32 healthy controls.

**Figure 4 tropicalmed-09-00252-f004:**
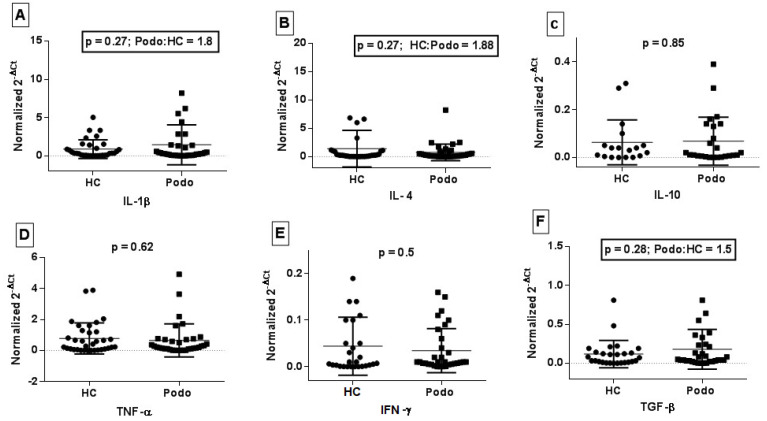
Scatter plots showing comparisons of the average expression of mRNA for six cytokine genes for podoconiosis cases and healthy controls. Plot (**A**–**F**) shows normalized gene expression data for *IL-1β*, *IL-4*, *IL-10*, *TNF-α*, *IFN-γ*, and *TGF-β*, respectively. The horizontal line in the middle represents the median from the 2^−∆Ct^ value; the two lines above and below this indicate interquartile ranges for 38 podoconiosis (Podo) patients and 32 healthy control (HC) subjects.

**Figure 5 tropicalmed-09-00252-f005:**
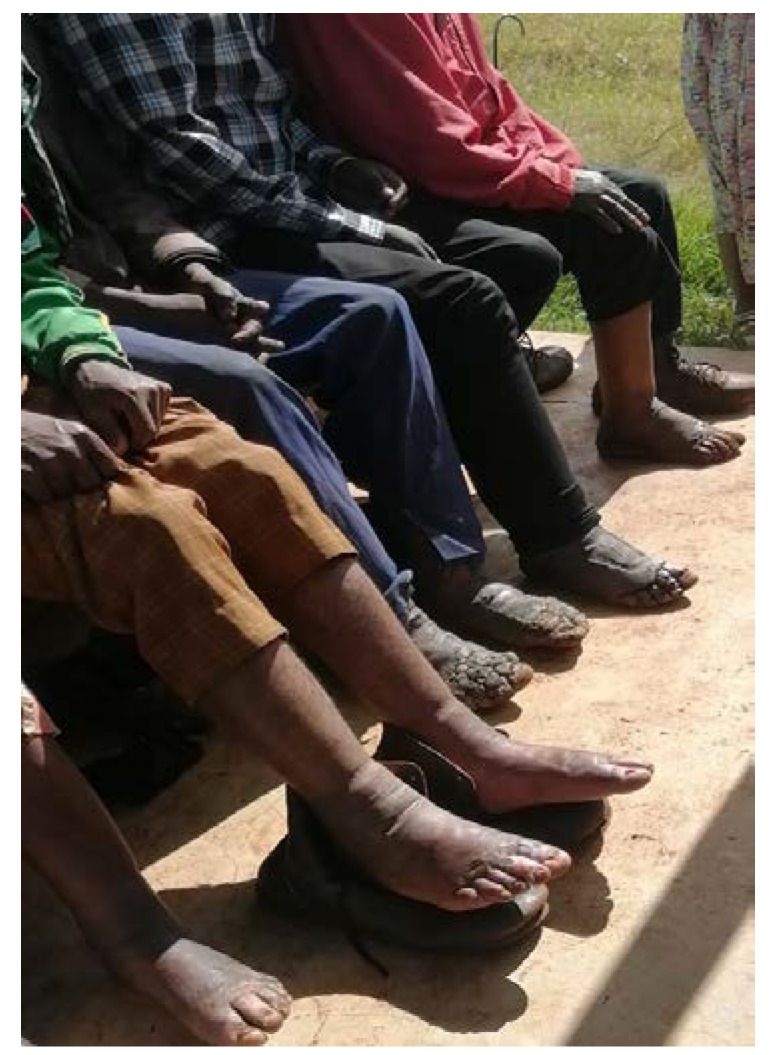
A photo of selected podoconiosis patients taken from one of the study areas!

**Table 1 tropicalmed-09-00252-t001:** Socio-demographic characteristics of podoconiosis patients and healthy controls in North West Ethiopia.

Variable	Podoconiosis (%), *n* = 56	Healthy Control (%) *n* = 44
**Sex:** Male Female	32 (57)	26 (59)
	24 (43)	18 (31)
**Age (years):** Mean (SD)	45 (10.3)	37 (9.3)
**Occupation:** Farmer Student	56 (100)	40 (91) 4 (9)
**Marital status:** Married Divorced Single	47 (84) 7 (12.5) 2 (3.5)	35 (79.5) 4 (9) 5 (11.5)
**Disease stage:** Stage 2 Stage 3	52 (92.8) 4 (7.2)	
**Duration of disease** **in years:** mean (sd)	17 (9.2)	
**Site involved:** Both legs Left leg Right leg	50 (89.3) 4 (7.2) 2 (3.5)	

**Table 2 tropicalmed-09-00252-t002:** Time line for measurement of *TNF-α*, *IL-1β*, and *IFN-γ* levels by ELISA in culture supernatants following 24 h and 6 days of stimulation under different conditions.

Cytokine	Day 1	Day 2	Day 3	Day 6
*TNF-α* and *IL-1β*	Minerals added to PBMC	SN harvested for ELISA	--	--
*IFN-γ*	PHA added to prime PBMCs	Minerals added to PBMCs	IL-2 added	SN harvested for ELISA

*TNF-α*; tumor necrosis factor alpha, *IL-1β*; interleukin 1-beta, *IFN-γ*; interferon-gamma, PBMC; peripheral blood mononuclear cells, SN; supernatant, PHA; phytohaemaglutinin.

**Table 3 tropicalmed-09-00252-t003:** List of primers used for qRT-PCR.

S.No.	Primer Name	Sequence	Tm	%GC
1	HuPO-Fw	GCTTCCTGGAGGGTGTCC	60.5	67
	HuPO-Rv	GGACTCGTTTGTACCCGTTG	59.4	55
2	*IL-4*-Fw	GGCAGTTCTACAGCCACCAT	59.4	55
	*IL-4*-Rv	TGGTTGGCTTCCTTCACAGG	59.4	55
3	*IFN-γ*-Fw	GGCTTTTCAGCTCTGCATCC	59.4	55
	*IFN-γ* Rv	TCTGTCACTCTCCTCTTTCCA	58	48
4	*IL-10* Fw	TGAGAACCAAGACCCAGACA	57	50
	*IL-10* Rv	TCATGGCTTTGTAGATGCCT	55	45
5	*TNF-α* Fw	AGCCCATGTTGTAGCAAACC	57	50
	*TNF-α* Rv	GCTGGTTATCTCTCAGCTCCA	60	52
6	IL-1 β Fw	AGCCCAGCCAACTCAATTC	57	53
	IL-1 β Rv	CATGGAGAACACCACTTGTTGC	60	50
7	*TGF-β*-Fw	GGACCAGTGGGGAACACTAC	61	60
	*TGF-β*-Rv	TAAAGCAGGTTCCTGGTGGG	59	55

HuPO—human acidic ribosomal phosphate, IL—interleukin, TNF—tumor necrosis factor, TGF—transforming growth factor, Fw—forward primer, Rv—reverse primer.

## Data Availability

All available data are included in the manuscript and the Appendix A. Datasets available on request from the authors.

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
