# Peer review of "Differences in Cytokine Expression at Baseline and in Response to Mineral Stimulation by Peripheral Blood Mononuclear Cells from Podoconiosis Cases and Healthy Control Individuals"

_tropicalmed, 2024, doi:10.3390/tropicalmed9110252_

Round 1
Reviewer 1 Report
Comments and Suggestions for Authors
This study evaluated inflammatory cytokine release from PBMCs of individuals in Northwest Ethiopia with (n= 56) and without (n=44) podoconiosis. Minerals tested were kaolinite, chlorite, and beryllium sulfate.
The authors found that spontaneous release of both TNFalpha and IL1beta was greater from PBMCs of individuals with podoconiosis than from PBMCs of healthy controls.
Additionally, the authors found that kaolinite and chlorite induced significantly greater release of IL1B from PBMCs of individuals with podoconiosis than from PBMCs of healthy controls. TNFalpha release was not significantly different in response to mineral stimulation between patients and controls.
Finally, the authors found that incubation of PBMCs with these minerals tended to decrease inflammatory cytokine release in long term culture after initial stimulation with PHA, but that the decreases were greater in healthy controls than for PBMCs of patients with podoconiosis.
The findings demonstrate that at baseline PBMCs of individuals with podoconiosis release greater amounts of inflammatory cytokines than healthy controls, and that minerals found in volcanic soil (especially kaolinite and chlorite) can induce further increased release of IL1beta from indviduals with podoconiosis.
The study was fairly well written, the research design reasonable, and the statistical analyses appropriate. The large number of patients with podoconiosis evaluated is a strength. Overall, the general conclusions appear reasonable though a couple of specific conclusions (as mentioned in critiques 1 and 2 listed below) do not directly align with the results. The discussion could be improved by mentioning the possible mechanisms or cell types that may be involved in the production of inflammatory cytokines in reponse to volcanic minerals (see critique 3 below).
Critiques
- The title states “Enhanced expression of pro-inflammatory cytokines in peripheral blood….” As far as I can tell, no results are presented of cytokine concentrations in peripheral blood. The title should be changed to something like “Peripheral blood mononuclear cells from individuals with podoconiosis exhibit greater production of pro-inflammatory cytokines at baseline and in response to mineral stimulation”
- In the abstract, the authors write that “The minerals induced higher levels of these two cytokines following 24hrs of stimulation in both study groups.” Data and text for Figure 1 demonstrate only enhanced IL1B in response to kaolinite and chlorite. TNF alpha release from PBMCs of podoconiosis patients is not greater for any of the minerals tested.
- The general finding that PBMCs exhibit increased production of some inflammatory cytokines when minerals are directly incubated with them is interesting. The authors repeatedly mention that there are MHCII links with podoconiosis….but it is not clear how minerals would activate CD4 T cells (which typically responde only to peptides presented in the context of MHCII. In their discussion, the author should mention the possible mechanisms by which minerals may be directly activating PBMCs to release pro-inflammatory cytokines. They should also discuss which cells may be involved.
The paper is generally well written.
Author Response
Critiques
- The title states “Enhanced expression of pro-inflammatory cytokines in peripheral blood….” As far as I can tell, no results are presented of cytokine concentrations in peripheral blood. The title should be changed to something like “Peripheral blood mononuclear cells from individuals with podoconiosis exhibit greater production of pro-inflammatory cytokines at baseline and in response to mineral stimulation”
Thank you for your overall comment and suggestion and we agree and have edited the topic per your suggestion.
2. In the abstract, the authors write that “The minerals induced higher levels of these two cytokines following 24hrs of stimulation in both study groups.” Data and text for Figure 1 demonstrate only enhanced IL1B in response to kaolinite and chlorite. TNF alpha release from PBMCs of podoconiosis patients is not greater for any of the minerals tested.
Thanks for pointing this out, we have modified the statement on the abstract line 10 to show that the two minerals induced the response on IL1B following 24hrs stimulation.
3. The general finding that PBMCs exhibit increased production of some inflammatory cytokines when minerals are directly incubated with them is interesting. The authors repeatedly mention that there are MHCII links with podoconiosis….but it is not clear how minerals would activate CD4 T cells (which typically responde only to peptides presented in the context of MHCII. In their discussion, the author should mention the possible mechanisms by which minerals may be directly activating PBMCs to release pro-inflammatory cytokines. They should also discuss which cells may be involved.
We agree with this suggestion and we have added a paragraph which shows a potential mechanism for interaction of minerals with MHC molecules with berylliousis model as an example (this is highlighted in the last paragraph of the discussion. We have also tried to mention that the inflammatory cytokines are likely induced from myeloid cells as we have observed higher activation markers in monocytes and DC from previous experiments in the our lab (Paragraph 5 line 11-14)
Reviewer 2 Report
Comments and Suggestions for Authors
General comment
This study provides valuable data on a common clinical case in Africa and some other regions of the worlds on podoconiosis. This data can be exploited in control or prevention of such cases. However, I strongly recommend the author to not use the expression of revealing pathogenesis because this is a kind of study that provides clinical cases data while pathogenesis or mechanisms studies required highly advanced and accurate laboratory procedures. Also, some points need clarification by the authors before accepting the manuscript as will be reported in below.
Abstract
Novelty and aspect of application of obtained knowledge should be clearly described in the abstract.
Introduction
- Why did the authors focused on testing TNF-α, IL-1β, and IFN-É£ using ELISA? They should add relevant information in the introduction and discussion.
Results
The authors should divide figure 2 into figure 2 for panels A, B, and C, and figure 3 for panels D, E, and F and correspondingly figure 3 changed to figure 4. In addition, I strongly ask the authors to add each figure in the correct location in the manuscript after relevant citation.
Discussion
- Line 202, change style of “in vivo” to italic style and confirm the same notes for “in vivo, in vitro or Latin expressions” throughout the manuscript.
- Lines 238-253, change cytokine gene names to italic style and confirm the same notes throughout the manuscript.
Materials and methods
- The authors should clearly describe the criteria for dividing groups into podoconiosis and healthy and how they define or diagnose the members of each group?
- It would be valuable if the authors can add relevant photos of the patients.
- Lines 295-317, the citation or validation criteria are needed for the used procedures of ELISA that based on multiple time testings. Also, how the authors detected the dose of used minerals in vitro and did they check any parameters for cytotoxicity measurements or changes in the incubated cells?
Comments on the Quality of English LanguageMinor issues are detected and described in the comments to authors section.
Author Response
1.Novelty and aspect of application of obtained knowledge should be clearly described in the abstract.
We have tried to highlight the main finding of the result and we have tried to tone down the pathogenesis or mechanism description and tried to highlight it as a gap to be addressed in future more succinate studies.
Introduction
- Why did the authors focused on testing TNF-α, IL-1β, and IFN-É£ using ELISA? They should add relevant information in the introduction and discussion.
The minerals are non specific stimulants and in different mechanism they may induce cytokine production from myeloid cells (pro-inflammatory cytokines through NLP3 pathway-described in paragraph five of the discussion and INF-y through crosslinking or binding with the HLA molecule, a paragraph describing this with berylleosis model is now added in the last paragraph of the discussion
Results
The authors should divide figure 2 into figure 2 for panels A, B, and C, and figure 3 for panels D, E, and F and correspondingly figure 3 changed to figure 4. In addition, I strongly ask the authors to add each figure in the correct location in the manuscript after relevant citation.
We agree with this comment and we have edited the figures in to two figures of Figure 2 and 3, then cited in relevant places.
Discussion
- Line 202, change style of “in vivo” to italic style and confirm the same notes for “in vivo, in vitro or Latin expressions” throughout the manuscript.
We have modified to italic form the “in vivo” and “in vitro” terms
- Lines 238-253, change cytokine gene names to italic style and confirm the same notes throughout the manuscript.
Cytokine gene names are changed to italic font throughout the manuscript
Materials and methods
- The authors should clearly describe the criteria for dividing groups into podoconiosis and healthy and how they define or diagnose the members of each group?
We have now indicated this more in the methods of study subjects section from line 10-16. We indicated that the patients were already diagnosed and staged based on Tekola staging system (reference cited). The patients had morbidity management follow up and are already recorded in dedicated chart. Control selection criteria is also described more.
- It would be valuable if the authors can add relevant photos of the patients.
A picture taken from one of the study areas without showing the face of the participants is now included in the study subjects section of the methods
- Lines 295-317, the citation or validation criteria are needed for the used procedures of ELISA that based on multiple time testings. Also, how the authors detected the dose of used minerals in vitro and did they check any parameters for cytotoxicity measurements or changes in the incubated cells?
The ELISA procedures is carried out based on the manufacturer instruction and also the procedure is carried out routinely in our lab. Prior to sample testing reproducibility and accuracy were checked with serial dilution of the standards provided.
The dose of used minerals were tested in optimization experiments with serial dilution and the final concentration used was the least cytotoxic dose for the cells. Moreover, this dose was used in prior silica based in vitro experiments. This is now indicated in in vitro mineral stimulation sub section line 10-13.
Comments on the Quality of English Language
Minor issues are detected and described in the comments to authors section.
We have gone through the manuscript to edit the English and some typos error
Round 2
Reviewer 2 Report
Comments and Suggestions for Authors
The authors responded to comments and the manuscript has been improved accordingly.